# Use of Three Different Nanoparticles to Reduce Cd Availability in Soils: Effects on Germination and Early Growth of *Sinapis alba* L.

**DOI:** 10.3390/plants12040801

**Published:** 2023-02-10

**Authors:** Rocío González-Feijoo, Andrés Rodríguez-Seijo, David Fernández-Calviño, Manuel Arias-Estévez, Daniel Arenas-Lago

**Affiliations:** Department of Plant Biology and Soil Science, Área de Edafoloxía e Química Agrícola, Facultade de Ciencias, Universidade de Vigo, 32004 Ourense, Spain

**Keywords:** cadmium, nanoparticles, germination, seedling growth, *Sinapis alba* L.

## Abstract

Globally, cadmium (Cd) is one of the metals that causes the most significant problems of contamination in agricultural soils and toxicity in living organisms. In this study, the ability of three different nanoparticles (dose 3% *w*/*w*) (hydroxyapatite (HANPs), maghemite (MNPs), or zero-valent iron (FeNPs)) to decrease the availability of Cd in artificially contaminated agricultural soil was investigated. The effect of Cd and nanoparticles on germination and early growth of *Sinapis alba* L. was also assessed by tolerance/toxicity bioassays. The available Cd contents in the contaminated soil decreased after treatment with the nanoparticles (available Cd decreased with HANPs: >96.9%, MNPs: >91.9%, FeNPs: >94%), indicating that these nanoparticles are highly efficient for the fixation of available Cd. The toxicity/tolerance bioassays showed different behavior for each nanoparticle. The HANPs negatively affected germination (G(%): 20% worsening compared to control soil), early root growth (G_index_: −27.7% compared to control soil), and aerial parts (Ap_index_: −12%) of *S. alba*, but showed positive effects compared to Cd-contaminated soils (G_index_: +8–11%; Ap_index_: +26–47%). MNP treatment in Cd-contaminated soils had a positive effect on germination (G(%): 6–10% improvement) and early growth of roots (G_index_: +16%) and aerial parts (Ap_index_: +16–19%). The FeNPs had a positive influence on germination (G(%): +10%) and growth of aerial parts (Ap_index_: +12–16%) but not on early growth of roots (G_index_: 0%). These nanoparticles can be used to reduce highly available Cd contents in contaminated soils, but MNPs and FeNPs showed the most favorable effects on the early growth and germination of *S. alba*.

## 1. Introduction

One of society’s most critical environmental issues in the twenty-first century is soil loss and degradation. Around the world, there are considerable pollution issues and soil deterioration due to the excess of potentially harmful compounds of different natures integrated into the soil each year [1,2,3]. These alterations may risk soil health, human health, and other living organisms [4]. Consequently, soil pollution must be assessed, in order to limit other adverse effects, such as destruction of the soil’s self-purification capacity; qualitative and quantitative decreases in soil microorganism growth; a decrease in crop yields with potential changes in end-product properties; or contamination of surface and phreatic waters by transfer processes.

Metals may cause soil pollution problems since they may undergo significant changes in their soil contents due to anthropogenic activities, such as agriculture, mining, industrial processes, and urban planning [5]. The hazard of metals lies in the fact that they cannot be degraded and may tend to bioaccumulate and biomagnify in living organisms, and may cause toxic effects [6]. The metal persistence in soils can be greater than in other environmental compartments, and soil pollution by metals can often be considered to be almost permanent if appropriate measures are not taken. Naturally, metals accumulated in soils can be slowly removed by leaching, uptake by plants or other organisms, erosion, or deflation [7]. However, most metal-contaminated soils are linked to human activities, which usually have poor texture, lack of structure, low clay and organic matter contents, acidic pH, and low cation exchange capacity [8]. These soil characteristics also limit plant growth since they favor high metal mobility and bioavailability, which is usually linked to an increase in toxicity for plants and other soil organisms, as well as to a higher transfer of these metals to different compartments of the ecosystem [9].

Agriculture is one of the activities that causes major metal contamination problems in soils. The extensive use of phytosanitary products, as well as irrigation with wastewater or water contaminated with high concentrations of metals, can cause issues of metal accumulation in crops and favor their incorporation into the trophic chain [10]. However, although the toxic effects of metals in soils are usually associated with the total metal concentration in the soil, in reality, it is the bioavailable fraction that is most important in the processes of uptake and accumulation in plants [9,11]. Thus, determining this fraction in the soil is highly relevant to assessing the toxicity of a given metal and its potential soil and environmental risks. For this, the assessment of the metal bioavailability in soils can usually be performed by methodologies that are based mainly on the use of single extractions [12], which can be used to determine the available metal content in the soil.

Furthermore, bioassays with plants can also be used to study the effects and possible toxicity caused by metals and/or their accumulation [13,14,15]. In addition, the ability of some plants to absorb metals enables the assessment of metal bioavailability. Establishing a vegetative cover that protects against soil erosion, runoff, and pollution in surrounding areas is an effective tool for recovering contaminated soils [16].

Cadmium is a metal found at low concentrations in nature and is usually present in rocks, water, soils, or even living beings. This metal has no essential biological function, although certain concentrations can cause toxicity [17]. Although high levels of Cd in soils may have a lithogenic origin, the primary sources of soil pollution with Cd are the manufacture of nickel-cadmium batteries, iron and steel production processes, and coal and oil industries [7]. Cadmium particles may also be frequently present in the air, making atmospheric deposition one of the main routes of soil contamination. In addition, a large amount of industrial wastewater from Zn and phosphate production contains Cd, and can be a source of soil pollution [17]. Soil pollution by Cd causes serious environmental problems, mainly in acidic soils that favor Cd fixation by plants, or soils vulnerable to acidification and with a shallow water table. More specifically, the presence of high Cd contents in many agricultural soils worldwide, as a consequence of irrigation with contaminated wastewater from industrial or mining discharges, has generated significant crop losses and severe health problems for humans [18,19,20].

In general, the Cd solubility increases with decreasing soil pH. Moreover, organic matter strongly adsorbs this metal, and plays a critical role in retaining and accumulating Cd in the soil. Similarly, the Cd concentration in soil solution, bioavailability, and toxicity are controlled by the pH, organic matter, cation exchange capacity, clay content, and Fe and Mn oxyhydroxides [21]. In addition, Cd in soils tends to accumulate in plants and can be incorporated into the food chain, via soil organisms susceptible to Cd [22,23]. Although Cd is considered a non-essential element for metabolic processes in plants, this element usually accumulates in roots and leaves. There is evidence that a considerable fraction of Cd can be passively absorbed by roots, although it is generally incorporated into the metabolic pathways of plants, causing toxicity problems [22]. Cadmium also interferes with the absorption, transport, and use of water and essential elements (Ca, Mg, P, and K), inducing water and nutritional imbalances in plants. Plants exposed to Cd-contaminated soils usually show changes in stomatal aperture, photosynthesis, and transpiration [24,25]. Another common symptom caused by Cd toxicity is chlorosis caused by Fe deficiencies, phosphates, or reduced Mn transport [26].

In recent years, there has been increasing interest in the application of nanoparticles for the study and potential remediation of environmental contaminants [27,28,29,30,31], especially aromatic compounds, dyes, pesticides, biological wastes, halogenated species, and metals [32]. Nano-remediation has emerged as a possible solution to remediate contaminated soils [33,34]. The higher efficiency and remediation rate of nano-amendments applied to contaminated soils compared to traditional amendments are based on a higher specific surface area and reactivity [35]. Nonetheless, the incorporation of these nanomaterials into the environment, especially soils, must be evaluated due to possible toxic effects on soil organisms. For this evaluation, it may be possible to carry out toxicity/tolerance bioassays, which, as with metals, also allow assessment of the effects of nanoparticles on growth and germination in reference plants.

Among the different nanomaterials proposed as potential remediating compounds for metal-contaminated soils, studies based on nanophosphates have provided satisfactory results [36,37]. In particular, hydroxyapatite nanoparticles can act in soils with high metal levels due to their high reactivity, small size, large specific surface area, and ease of spreading in the soil [36,38,39]. Moreover, Fe oxides and zero-valent iron nanoparticles show characteristics such as high reactivity and specific surface area, which can favor the immobilization of metal ions [28,40]. Specifically, maghemite, magnetite, and zero-valent iron nanoparticles have been successfully used to remove metals and metalloids from water and soil [41,42,43,44].

Therefore, the main objective of this work was to determine if HANPs, MNPs, or FeNPs together with a plant species such as *S. alba* can be effective for assessing Cd-contaminated soils and improving the conditions of agricultural soils contaminated with this metal. The specific objectives were: (i) to determine the availability of Cd in artificially contaminated agricultural soil; (ii) to assess the toxicity of Cd in *S. alba* as a consequence of the addition of Cd and different nanoparticles to the soil; and (iii) to study the effects of the nanoparticles on the germination and early growth of *S. alba* by toxicity/tolerance bioassays.

## 2. Results and Discussion

### 2.1. Soil Characterization

The general soil characteristics are shown in Table 1.

The selected agricultural soil shows a sandy loam texture, which indicates that it has a high sand content and lower silt and clay content. Clays are the mineralogical components that most influence metal retention [45]. This soil, with low clay content, is suitable for studying the effect of different nanoparticles in Cd retention, minimizing the possible interactions between Cd and clay particles.

This soil has a moderately acidic pH. pH is one of the properties that most affects the retention of Cd in soils. In general, pH is controlled by soil components such as clay minerals, organic matter, aluminum and iron oxides, and the parent material. At low pH, H^+^ is strongly retained on particle surfaces, which favors a higher solubility of Cd, an increase in its concentration in the soil solution, and thus a higher bioavailability [17]. In addition, Al^3+^ is the predominant cation in the exchange complex in this soil, which may favor lower pH if aluminum becomes available. The soil pH_(KCl)_ is very strongly acidic.

The soil has a low organic matter content (2%), so its effect on Cd retention and the studied nanoparticles is relatively low. However, organic matter can play an essential role in Cd fixation and interaction with the nanoparticles through ion exchange reactions, surface adsorption, chelation, coagulation, and/or peptization [7,46]. In this study, we tried to select a soil with low organic matter content to minimize these interactions and to evaluate more effectively the effect of nanoparticles on Cd, but without eliminating its presence, since the organic matter content in soils around the world is between 2 and 10% [47].

The ECEC of the soil is high due to the high Al^3+^ content in the exchange complex. In contrast, the contents of basic cations in the exchange complex are very low, mainly for Ca^2+^, Mg^2+^, and Na^+^.

The soil has relatively high contents of Fe oxides, while those of Al and Mn are much lower. The Fe and Mn oxides have an important influence on the retention of metals in soils through surface chemisorption processes [48].

The pseudo-total metal contents in the soil are also shown in Table 1. It should be noted that the Cd content in the soil is below the detection limit of ICP-OES (<0.01 mg kg^−1^). Thus, adding Cd solutions is not affected by the presence of naturally occurring Cd in the soil. Regarding the rest of the metals, no element contents are above the generic reference levels for phytotoxicity [49].

### 2.2. Available Cd Contents in Soil Samples

The available Cd contents (extracted with CaCl_2_) in the untreated and treated soil samples with Cd and/or nanoparticles are shown in Table 2. Moreover, the soil pH_(H2O)_ after each treatment is shown in Table 2.

No available Cd was detected in the untreated soil since, as indicated above, the total Cd content in the soil is below the detection limit of ICP-OES (<0.01 mg kg^−1^). The available Cd contents in the soils treated with the Cd solution (1 and 3 mg kg^−1^) (without nanoparticles) were lower relative to the initially added Cd contents, with available concentrations of 0.5 mg kg^−1^ (added solution of 1 mg kg^−1^) and 1.6 mg kg^−1^ (added dilution of 3 mg kg^−1^) of available Cd in the soil samples. These results indicate that approximately 50% of Cd is available in the soil after treatment. The other 50% of Cd may be retained by soil components such as organic matter or oxides.

The available Cd contents in soil samples treated only with nanoparticles (without added Cd) also showed no available Cd in soils. In soil samples with Cd added + HANPs, the available Cd content decreased by more than 98% (1 mg kg^−1^ Cd treatment) and 96.9% (3 mg kg^−1^ Cd treatment) relative to the available Cd contents in soils without nanoparticles. In agreement with these results, different authors have indicated that phosphate-based nanoparticles can decrease Cd availability and favor its immobilization in soils. For example, HANPs combined with fungi may favor Cd retention through metal immobilization mechanisms, including precipitation of new metal phosphate minerals [50]. Similarly, synthesized biochar-supported iron phosphate nanoparticles may remediate Cd-contaminated soils, with immobilization efficiencies of 81 and 60% of available Cd, respectively [51,52]. Moreover, iron phosphate nanoparticles were used in previous studies to reduce Cd leaching in soils. These nanoparticles were highly effective at immobilizing exchangeable Cd and increasing the residual fraction. This suggests that Cd phosphate precipitation is responsible for the decrease in Cd availability in soils [37]. In addition, in previous studies, HANPs and tricalcium phosphate nanoparticles also effectively decreased the content of elements, such as Cd, As, Cu, Pb, Sb, and Zn in mine soils [34] and firing range soils [31,33,36].

Similar results were found in soil samples treated with Cd and MNPs. After adding these nanoparticles, the available Cd content decreased by 94% (1 mg kg^−1^ Cd treatment) and 91.9% (3 mg kg^−1^ Cd treatment) relative to soil samples treated with Cd but without nanoparticles. Similarly, in soil samples with FeNPs, the available Cd content also decreased by more than 98% (1 mg kg^−1^ Cd treatment) and 94% (3 mg kg^−1^ Cd treatment) compared to soil samples with Cd but without nanoparticles. In this context, several authors have indicated that using maghemite nanoparticles can effectively reduce Cd availability and favor its immobilization in the soil. For example, silica-coated maghemite nanoparticles can extract metals from soil dissolution, showing high immobilization efficiency (>70%) [53]. Moreover, maghemite nanoparticles effectively reduced the concentration of As and Cd in soil dissolution [54] and available As, Pb, and Sb in mine soils [34]. Other studies have indicated that zero-valent iron nanoparticles also have a high capacity to immobilize Cd in soil. For example, zero-valent iron nanoparticles can be used to immobilize Cd, Cr, and Pb in soils contaminated with these metals, favoring the growth of *Brassica chinesis* L. [55]. Similarly, zero-valent iron nanoparticles can reduce the Cd and Pb bioavailability in artificially contaminated soils and enhance the growth of *Acer velutinum* Bioss. [43].

In addition, soil pH was determined after treatment with Cd and/or nanoparticles (Table 2). The results indicated that adding HANPs increased soil pH in both untreated and Cd-treated soils, which may be another determining factor favoring Cd retention. Regarding the pH values after treatment with the MNPs and FeNPs, they generally increased with both nanoparticles, but mainly upon the addition of the FeNPs, which, as indicated above, may also be a factor favoring Cd retention.

### 2.3. Toxicity/Tolerance Bioassays with S. alba

Toxicity/tolerance bioassays with *S. alba* seeds (in untreated soil and soil treated with Cd and/or nanoparticles) were carried out to evaluate the toxicity of Cd and the effects of the different nanoparticles on germination and early growth of this species. Examples of the plates with germinated seeds for each treatment are shown in Figure 1. Moreover, the percentage of germinated seeds G (%), root and aerial part length, and G_Index_ are shown in Table 3.

The results show high germination of *S. alba* seeds in both untreated and treated soil samples (≥80%). Considering the untreated soil samples as a control, there were no significant differences in G (%) between the control samples and those treated with MNPs and FeNPs (without added Cd). In this context, a study showed the possible toxicity of metal oxide (including Fe_2_O_3_) nanoparticles on the growth of *S. alba* without finding adverse effects on germination [56]. In contrast, G (%) decreased slightly relative to untreated soil samples when treated with HANPs (Table 3).

In Cd-treated soil samples, the G (%) of *S. alba* seeds decreased relative to the untreated soil samples. In soil samples treated with HANPs and Cd, despite the slight negative effect of these nanoparticles on germination in the soil without Cd (G (%): 80%), the G (%) increased in those treated with Cd and HANPs (G (%): 86–90%). This result suggests a positive effect of HANPs on germination when high levels of Cd are present in the soil. Conversely, there were no significant differences in G (%) between the soil sample controls and the soil samples treated with MNPs or FeNPs. In contrast, the G (%) improved when MNPs (6–10%) or FeNPs (10%) were added to the Cd-treated soil samples. This result suggests that MNPs and FeNPs could be used to decrease available Cd contents in contaminated soils and to enhance the germination of *S. alba* seeds in Cd-contaminated soils.

To evaluate the Cd toxicity and the effect of nanoparticles on the roots’ and aerial parts’ early growth, G_Index_ and Ap_Index_ were calculated (Table 3). The results obtained from G_Index_ indicated that the addition of the nanoparticles to the soil samples (without Cd), in general, did not favor the early root growth of *S. alba* relative to the control. The G_Index_ values < 90% indicated an inhibitory effect on early root growth when nanoparticles are added to the soil samples. However, adding HANPs and MNPs considerably improved G_Index_ (HANPs: 8–11%; MNPs: +16%) for Cd-contaminated soil samples (both 1 and 3 mg kg^−1^). For FeNPs, no differences were observed between Cd-contaminated soil samples and soil samples treated with these nanoparticles. Thus, although G_Index_ in the control soil samples is the highest, the results suggest that, with high Cd contents, the Cd toxicity on root early growth in *S. alba* could be mitigated by adding HANPs and MNPs.

The results for Ap_Index_ indicated that the addition of nanoparticles to the soil samples without Cd treatment, in general, did not favor the growth of the aerial part of *S. alba* regarding the control soil samples (HANPs: 87.3; MNPs: 93.3; FeNPs: 89.1). In any case, regarding the Cd-treated soil samples (both and 3 mg kg^−1^), the addition of nanoparticles improved the Ap_Index_ values (HANPs: 26–47%; MNPs: 16–19%; FeNPs: 12–16%). Therefore, MNPs added to Cd-treated soil samples positively affected germination, G_Index_, and Ap_Index_ of *S. alba* L.

Germination inhibition (GI) percentages and root growth inhibition (RI) percentage data were calculated from germination percentage and root growth data (Table 3) and are shown in Table 4.

The results indicated that there was inhibition of germination (GI) and root inhibition (RI) (relative to the control soil) when soils were treated with HANPs. This indicates that HANPs can negatively affect the germination and early root growth of *S. alba*. In contrast, there was no GI in *S. alba* seeds in soil samples treated with MNPs and FeNPs compared to control soil samples, although RI was observed. Therefore, all nanoparticles had a negative effect on the early root growth of *S. alba* in soils without Cd contamination.

By comparison, there were GI and RI (relative to the control soil) in soils treated with Cd (without nanoparticles). After nanoparticle treatments, different behaviors were observed. In soils with Cd and HANPs, RI decreased relative to soils with Cd. In contrast, GI only decreased for soils with Cd (1 mg kg^−1^) and HANPs. In soil samples treated with MNPs and FeNPs, GI decreased considerably, limiting germination problems caused by high Cd contents. In this context, RI also decreased with MNPs but increased with FeNPs. These results indicate that HANPs did not positively affect germination and early root growth of *S. alba* in Cd-contaminated soils. The MNPs had a positive impact on germination and early growth of roots. The FeNPs had a positive effect on germination but not on the early growth of roots. 

## 3. Materials and Methods

### 3.1. Soil Sampling and Characterization

For this study, a natural agricultural soil classified as Cambic Umbrisols [57], which had not been managed and suffered anthropogenic alterations for the last 15 years, was selected (42° 8′ 9.17″ N 8° 46′ 34.37″ W Datum ETRS89—Nigrán, Pontevedra, Galicia, Spain). Using an Eijkelkamp sampler, six soil samples were taken from a depth of 0 to 30 cm (surface horizon) and stored in polyethylene bags. Soil samples were pooled, air dried, passed through a 2 mm sieve, and homogenized in a pooled soil sample. Then, the pooled soil sample was divided into a Fritsch Laborette 27 rotary sample divider in subsamples for soil analysis and Cd and/or nanoparticle soil treatments.

The soil was characterized exhaustively for pH in distilled H_2_O and KCl 0.1 M (soil water suspension ratio 1:2.5) [58]. Organic matter was determined following the Walkley and Black method [59]. The organic matter content was calculated by multiplying the total carbon content by the Van Bemmelen factor (1.724 for carbon contents <5.8%). Aluminum, Fe, and Mn oxides were determined using the dithionite-citrate method [60]. The contents of Fe, Al, and Mn oxides were measured by ICP-OES (Perkin Elmer Optima 4300DV, PerkinElmer Inc., Waltham, MA, USA) after the extraction method, and the results were expressed as Fe_2_O_3_, Al_2_O_3_, and MnO, respectively. Soil texture and mineral particle size analysis were determined by particle size analysis. Sand content was determined by wet sieving, and clay and silt contents were determined by pipette based on Stokes’ law [61]. The exchangeable basic cations (Ca^2+^, Mg^2+^, Na^+^, K^+^) were determined according to the Hendershot and Duquette method [62]. The exchange bases (Na^+^, K^+^, Mg^2+^, and Ca^2+^) were determined by ICP-OES (Perkin Elmer Optima 4300 DV). Aluminum (Al^3+^) in the exchange complex was determined by titration with NaOH (0.01 M). The effective cationic exchangeable capacity of the soil was determined as the sum of basic cations and Al^3+^. The results were expressed in cmol_(+)_ kg^−1^ of soil.

The pseudo-total metal content in the soil was determined by acid extraction in a microwave oven: 0.2 g of soil was weighed and added to a Teflon reactor, and 1 mL of HNO_3_ (65% *v*/*v*) and 3 mL of HCl (37% *v*/*v*) were also added. Then, the digestion was carried out in the microwave. Once the digestion process was completed, the sample was filtered and transferred to a 100 mL volumetric flask. The metal contents were determined by ICP-OES (Perkin Elmer Optima 4300 DV).

### 3.2. Nanoparticles

Three different nanoparticles were selected to assess the availability of Cd in the soil and the effects of both Cd and nanoparticles on the germination and early growth of *S. Alba* L.:

(i) Hydroxyapatite nanoparticles (HANPs) with a filamentous morphology (purity: 99%; nominal size: 20 nm). Reference: MKN-HXAP-020P Ca_10_(PO_4_)_6_(OH)_2_ supplied by mKnano (M K Impex Corp., Mississauga, ON, Canada).

(ii) Maghemite nanoparticles (MNPs) with a spherical morphology (purity: 99%; nominal size: 20–40 nm). Reference: NO-0053-HP-0500 supplied by IoLiTec nanomaterials (IoLiTec Liquids Technologies GMbH, Heilbronn, Germany).

(iii) Zero-valent iron nanoparticles (FeNPs) (nominal size: 25 nm). Reference: MKN-nZVI-S025 supplied by mKnano (M K Impex Corp., Mississauga, ON, Canada).

### 3.3. Plant Species

Seeds of *S. alba* were chosen for tolerance/toxicity bioassays. *S. alba* seeds were supplied by Herbiseed (Herbiseed, Twyford, Berkshire, UK). This species was selected because of its rapid growth, high metal tolerance, and its common use as a species in metal tolerance and toxicity bioassays.

### 3.4. Cadmium and/or Nanoparticle Soil Treatments

A quantity of 90 g of soil was placed in glass flasks, and the three different nanoparticles were added at 3% (*w*/*w*) (HANPs, MNPs, or FeNPs) (Figure 2). Soil samples that did not carry nanoparticles were omitted from the previous step. Then, the necessary volume of CdCl_2_ in an aqueous solution was added to meet two conditions (Table 5): (i) the final concentration of Cd in the soil samples was 1 or 3 mg kg^−1^ soil, and (ii) the soil samples were at field capacity. Subsequently, soil samples were shaken on a rotary shaker for 48 h. The available Cd contents in the soil samples were determined by ICP-OES (Perkin Elmer Optima 4300 DV). The Cd-contaminated soil samples were used for toxicity/tolerance bioassays with *S. alba.* The added Cd concentrations were selected and based on the maximum generic reference for ecosystem protection (1 mg kg^−1^) and phytotoxic levels (3 mg kg^−1^) [49].

### 3.5. Toxicity/Tolerance Bioassays Procedure

Toxicity/tolerance bioassays (provided by Phytotoxkit) were performed following the procedure recommended by MicroBioTests Inc. (MicroBioTests Toxkit) (Phytotoxkit, 2004) with *Sinapis alba* and the treatments indicated in Table 5 to determine the effect of Cd and/or nanoparticles on germination and early growth of *S. alba.* The bioassays were carried out on transparent test plates to observe and measure the seedling length (Figure 2). The toxicity/tolerance bioassays were conducted as follows:

(i) Soil samples (untreated and treated with Cd and/or nanoparticles) were placed at the bottom of the transparent plates (90 cm^3^) at their maximum water capacity. Subsequently, the soil was spread over each plate, flattened, and covered with a black filter (Figure 2A–D).

(ii) On each plate, 10 *S. alba* seeds were placed equidistantly on the black filter (Figure 2D). Then, the plates were covered, closed, and set upright. The plates were incubated for 3 days in the dark at 25 °C (Figure 2E). After incubation, seeds whose radicle was ≥1 mm were considered to be germinated and were recorded.

(iii) Finally, the plates were photographed with a digital camera (Figure 2F), and the images were analyzed with Image J software (version 1.52p—National Institutes of Health) to obtain root length and aerial part data. These data were used to determine and compare germination inhibition (%) and root growth inhibition (%) in all soil samples (treated and untreated with Cd and/or nanoparticles).

Germination inhibition (GI) and root growth inhibition (RI) were calculated according to the equation [13]:$$\mathrm{GI}\;o\;\mathrm{RI}\;\left(\%\right)=\frac{{\mathrm{SR}}_{c}-{\mathrm{SR}}_{s}}{{\mathrm{SR}}_{c}}\;\cdot \;100$$
where SR_C_ is the average seed germination or average root length for plants in the soil without Cd treatment and nanoparticles, and SR_S_ is the average seed germination or average root length in the different soils treated with Cd and/or nanoparticles.

In addition, the germination index (G_Index_) was used to evaluate Cd toxicity and nanoparticle effects on *S. alba.* G_Index_ was determined according to the equation [13]: $$G_{\mathrm{Index}}\left(\%\right)=\frac{G_{s}-L_{s}}{G_{c}\;\cdot \;L_{c}}\;\cdot \;100$$
where G_S_ and L_S_ are seed germination (%) and average root length (mm), respectively, in the different soils treated with Cd and/or nanoparticles, while G_C_ and L_C_ are their corresponding values in plants from the untreated soil. G_Index_ values were classified according to the following:G_Index_: between 90 and 110% is classified as a species without Cd toxicity.G_Index_: <90% is classified as a species with an inhibitory effect.G_Index_: >110% is classified as a species with a stimulant effect.

The influence index on the aerial part (P_Indice_) was determined from the equation:$${\mathrm{Ap}}_{\mathrm{Index}}\left(\%\right)=\frac{G_{s}-{\mathrm{Ap}}_{s}}{G_{c}\;\cdot \;{\mathrm{Ap}}_{c}}\;\cdot \;100$$
where G_S_ and Ap_S_ are seed germination (%) and average aerial part length (mm), respectively, in the different soils treated with Cd and/or nanoparticles, while G_C_ and P_C_ are their corresponding values in plants from the untreated soil.

It was considered that:Ap_Index_: between 90 and 110% is classified as a species without Cd toxicity on the aerial part.Ap_Index_: <90% is classified as a species with an inhibitory effect on the aerial part.Ap_Index_: >110% is classified as a species with a stimulating effect on the growth of the aerial part.

### 3.6. Determination of Available Cd Contents in Soil Samples

A simple extraction with CaCl_2_ (0.01 M) was carried out to determine the available Cd content in the soil samples (with and without nanoparticles) [63]. For this, 5 g of each soil sample (with and without nanoparticles) previously separated from the soil samples used in the toxicity/tolerance bioassays was weighed and stored in polypropylene tubes. After the same conditions used in toxicity/tolerance bioassays (3 days in the dark at 25 °C), 50 mL of a CaCl_2_ (0.01 M) solution was added. Then, each soil sample was shaken for 2 h, centrifuged, filtered, and analyzed for Cd content by ICP-OES (Perkin Elmer Optima 4300 DV). In conjunction with *S. alba*, the retention capacity of nanoparticles was estimated as the difference between the available Cd concentrations released from untreated soil samples and nanoparticle-treated samples (in aqueous extracts). The percentages of Cd retained in soils with and without nanoparticles were estimated for both Cd treatments (1 and 3 mg kg^−1^).

### 3.7. Statistical Analysis

All analyses, bioassays, and experiments were performed in triplicate. All data were statistically analyzed with IBM-SPSS v. 19.0 software for Windows. The mean and standard deviation of all measurements were determined. ANOVA and Duncan’s multiple range tests were used to compare differences between the different treatments with and without Cd and/or nanoparticles applied to the soil. Fisher’s minimum significant difference (LSD) test at 5% was used to compare means with weighted variance. The Kolmogorov–Smirnov and Levene tests were applied to verify the data normality and the homogeneity of the variances to check their homoscedasticity, respectively.

## 4. Conclusions

The available Cd content in the artificially contaminated soil decreased considerably after treatment with the different nanoparticles (HANPs, MNPs, and FeNPs), indicating that these nanoparticles could be used as highly available Cd fixers in similar Cd-contaminated agricultural soils. In addition, their ability as soil pH modifiers favors Cd retention. The Cd-treated soil samples significantly decreased the germination, and early growth of roots and aerial parts of *S. alba*. After treatment with MNPs and FeNPs, the germination improved considerably, and MNPs enhanced the early growth of roots and aerial parts of *S. alba*. However, HANPs slightly affected germination, while FeNPs affected the early growth of roots of *S. alba*. Therefore, the three nanoparticles can decrease Cd availability and prevent the transfer of this metal to other parts of the ecosystem. Nevertheless, only MNPs had no negative effects on *S. alba* compared to Cd-contaminated soils. This makes it necessary to further investigate the possible adverse effects of HANPs and FeNPs on the germination and early growth of *S. alba* and other species, mainly when they are used as potential environmental remediators.

## Figures and Tables

**Figure 1 plants-12-00801-f001:**
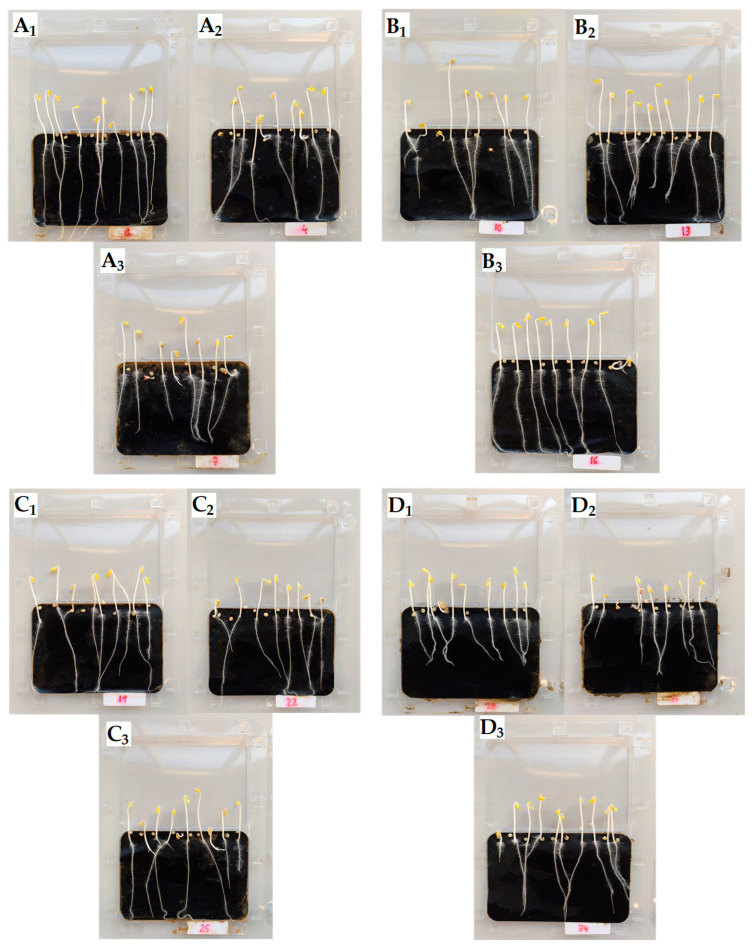
Toxicity/tolerance bioassays (examples) with *S. alba* for the control soil and different soil treatments: Without nanoparticles—(**A_1_**) Control soil, (**A_2_**) Soil + Cd (1 mg kg^−1^), (**A_3_**) Soil + Cd (3 mg kg^−1^); With HANPs (**B_1_**) Soil, (**B_2_**) Soil + Cd (1), (**B_3_**) Soil + Cd (3); With MNPs (**C_1_**) Soil; (**C_2_**) Soil + Cd (1); (**C_3_**) Soil + Cd (3); With FeNPs (**D_1_**) Soil; (**D_2_**) Soil + Cd (1); (**D_3_**) Soil + Cd (3).

**Figure 2 plants-12-00801-f002:**
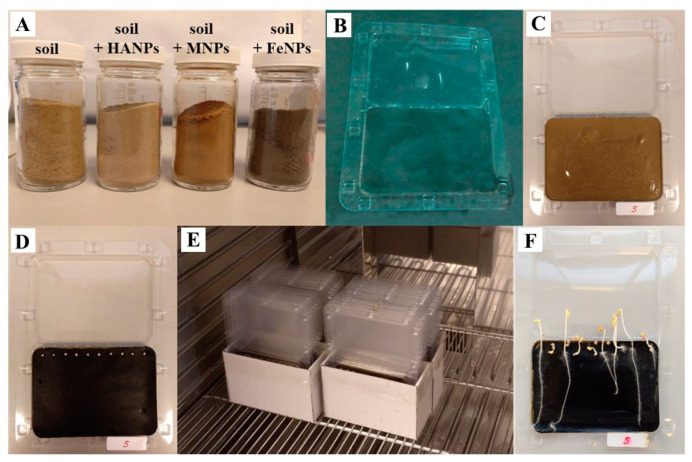
Stages of toxicity/tolerance bioassays (MicroBioTests Toxkit). (**A**) Soil samples with and without NPs; (**B**) Transparent plate; (**C**) Transparent plate with soil sample; (**D**) Transparent plate covered with a black filter and 10 seeds; (**E**) Transparent plate fit in the incubator camera; (**F**) Example of transparent plate with germinated seeds.

**Table 1 plants-12-00801-t001:** Soil physicochemical characteristics.

Characteristics	Unit	Value	Metal	Unit	Content
pH_(H2O)_	-	5.75 ± 0.10	Al	g kg^−1^	57.71 ± 8.69
pH_(KCl)_	-	4.85 ± 0.10	As	mg kg^−1^	14.58 ± 1.18
Organic matter	%	2.01 ± 0.35	Ca	g kg^−1^	1.76 ± 0.05
Texture	-	sandy-loam	Cd	mg kg^−1^	udl
			Co	mg kg^−1^	17.92 ± 0.38
Ca^2+^	cmol_(+)_kg^−1^	1.69 ± 0.34	Cr	mg kg^−1^	68.75 ± 2.18
K^+^		18.11 ± 4.84	Cu	mg kg^−1^	77.92 ± 1.84
Mg^2+^		0.50 ± 0.10	Fe	g kg^−1^	48.81 ± 3.08
Na^+^		1.50 ± 0.69	K	g kg^−1^	10.94 ± 1.07
Al^3+^		31.11 ± 1.97	Mg	g kg^−1^	10.88 ± 0.77
ECEC		52.92 ± 7.94	Mn	g kg^−1^	0.43 ± 0.02
			Na	mg kg^−1^	135.83 ± 3.54
Iron oxides	g kg^−1^	35.08 ± 4.17	Ni	mg kg^−1^	31.92 ± 1.26
Aluminum oxides		5.83 ± 0.70	Pb	mg kg^−1^	9.50 ± 1.32
Manganese oxides		0.15 ± 0.02	Zn	mg kg^−1^	107.67 ± 2.45

Exchangeable cations: Ca^2+^, K^+^, Mg^2+^, Na^+^, Al^3+^; ECEC: Effective Cation Exchange Capacity. udl: under detection limit.

**Table 2 plants-12-00801-t002:** Available Cd contents in untreated and treated soils with Cd and/or nanoparticles and pH after each treatment.

Soil + Cd Added Content (mg kg^−1^) + NPs	Available Cd (mg kg^−1^)	pH_(H2O)_
Control soil (untreated)	nd	5.75 ± 0.10 ^f^
Soil + Cd (1)	0.50 ± 0.08 ^b^	6.08 ± 0.03 ^d^
Soil + Cd (3)	1.60 ± 0.20 ^a^	5.94 ± 0.02 ^e^
Soil + HANPs	nd	6.32 ± 0.14 ^c^
Soil + Cd (1) + HANPs	<0.01 ^e^	6.32 ± 0.16 ^c^
Soil + Cd (3) + HANPs	0.05 ± 0.01 ^d^	6.42 ± 0.01 ^c^
Soil + MNPs	nd	6.01 ± 0.12 ^d^
Soil + Cd (1) + MNPs	0.03 ± 0.02 ^d^	6.05 ± 0.14 ^d,e^
Soil + Cd (3) + MNPs	0.13 ± 0.06 ^c^	6.12 ± 0.12 ^d^
Soil + FeNPs	nd	6.50 ± 0.24 ^c^
Soil + Cd (1) + FeNPs	<0.01 ^e^	6.92 ± 0.06 ^b^
Soil + Cd (3) + FeNPs	0.03 ± 0.02 ^d^	7.22 ± 0.08 ^a^

NPs: Nanoparticles, HANPs: Hydroxyapatite nanoparticles, MNPs: Maghemite nanoparticles, FeNPs: Zero-valent iron nanoparticles. Values with different letters differ significantly among treatments (*p* < 0.05). nd: not detected.

**Table 3 plants-12-00801-t003:** Germination percentage G (%), root length and aerial part (cm), germination index (G_Index_), and index of influence on the aerial part (Ap_Index_).

Soil + Cd Added Content (mg kg^−1^) + NPs	G (%)	G_Index_	Ap_Index_	Root Length	Aerial Part
Soil (untreated)	96 ± 4 ^a^	100.0 ± 0.6 ^a^	100.0 ± 1.0 ^b^	8.19 ± 2.11	4.16 ± 0.99
Soil + Cd (1)	80 ± 0 ^c^	60.9 ± 0.5 ^f^	60.9 ± 0.5 ^j^	5.98 ± 3.11	3.64 ± 1.14
Soil + Cd (3)	90 ± 0 ^b^	65.5 ± 0.4 ^e^	70.1 ± 0.5 ^i^	5.72 ± 2.02	3.11 ± 1.04
Soil + HANPs	80 ± 0 ^c^	63.3 ± 0.4	87.3 ± 0.6 ^f^	6.22 ± 1.84	4.36 ± 1.70
Soil + Cd (1) + HANPs	90 ± 5 ^a,b^	81.0 ± 0.5 ^c^	107.0 ± 1.0 ^a^	7.07 ± 1.82	4.76 ± 0.94
Soil + Cd (3) + HANPs	86 ± 6 ^b,c^	73.8 ± 0.6 ^d^	96.5 ± 0.6 ^c^	6.82 ± 2.96	4.48 ± 1.09
Soil + MNPs	96 ± 4 ^a^	82.9 ± 0.6 ^c^	93.3 ± 0.5 ^d^	6.74 ± 2.48	3.88 ± 0.81
Soil + Cd (1) + MNPs	90 ± 0 ^b^	87.0 ± 0.4 ^b^	80.4 ± 0.4 ^g^	7.60 ± 1.44	3.57 ± 0.84
Soil + Cd (3) + MNPs	96 ± 4 ^a^	81.5 ± 0.6 ^c^	86.2 ± 0.6 ^f^	6.72 ± 2.32	3.59 ± 1.19
Soil + FeNPs	96 ± 4 ^a^	56.5 ± 0.4 ^g^	89.1 ± 0.5 ^e^	4.62 ± 1.97	3.71 ± 0.96
Soil + Cd (1) + FeNPs	90 ± 0 ^b^	60.8 ± 0.5 ^f^	72.9 ± 0.4 ^h^	5.31 ± 2.40	3.24 ± 0.85
Soil + Cd (3) + FeNPs	100 ± 0 ^a^	64.8 ± 0.4 ^e^	86.0 ± 0.5 ^f^	5.09 ± 1.60	3.44 ± 0.90

NPs: Nanoparticles, HANPs: Hydroxyapatite nanoparticles, MNPs: Maghemite nanoparticles, FeNPs: Zero-valent iron nanoparticles. Values with different letters differ significantly among treatments (*p* < 0.05). nd: not detected.

**Table 4 plants-12-00801-t004:** Germination inhibition (GI) and root growth inhibition (RI) percentages.

Soil + Cd Added Content (mg kg^−1^) + NPs	GI (%)	RI (%)
Soil control (untreated)	0.00 ± 1.41 ^d^	0.00 ± 4.22 ^h^
Soil + Cd (1)	20.00 ± 0.70 ^a^	36.82 ± 5.22 ^d^
Soil + Cd (3)	6.67 ± 2.12 ^c^	43.09 ± 4.13 ^c^
Soil + HANPs	20.00 ± 0.70 ^a^	31.72 ± 3.95 ^d^
Soil + Cd (1) + HANPs	6.67 ± 2.12 ^c^	15.72 ± 3.93 ^f^
Soil + Cd (3) + HANPs	11.63 ± 1.41 ^b^	21.38 ± 5.07 ^e^
Soil + MNPs	0.00 ± 1.41 ^d^	20.63 ± 4.59 ^e^
Soil + Cd (1) + MNPs	6.67 ± 2.12 ^c^	7.78 ± 3.55 ^g^
Soil + Cd (3) + MNPs	0.00 ± 1.41 ^d^	22.71 ± 4.43 ^e^
Soil + FeNPs	0.00 ± 1.41 ^d^	77.04 ± 4.08 ^a^
Soil + Cd (1) + FeNPs	6.67 ± 0.71 ^c^	54.30 ± 4.51 ^b^
Soil + Cd (3) + FeNPs	−4.00 ± 0.71 ^e^	60.88 ± 3.71 ^b^

NPs: Nanoparticles, HANPs: Hydroxyapatite nanoparticles, MNPs: Maghemite nanoparticles, FeNPs: Zero-valent iron nanoparticles. Values with different letters differ significantly among treatments (*p* < 0.05). nd: not detected.

**Table 5 plants-12-00801-t005:** Treatments for the different soil analyses with Cd and/or nanoparticles.

Soil Samples	Cd Content Treatment	Nanoparticles Added (3% *w*/*w*)
90 g soil	-	-
	1 mg kg^−1^	-
	3 mg kg^−1^	-
90 g soil	-	HANPs
	1 mg kg^−1^	HANPs
	3 mg kg^−1^	HANPs
90 g soil	-	MNPs
	1 mg kg^−1^	MNPs
	3 mg kg^−1^	MNPs
90 g soil	-	FeNPs
	1 mg kg^−1^	FeNPs
	3 mg kg^−1^	FeNPs

## Data Availability

All data are shown in the manuscript.

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
