# Peer review of "Use of Three Different Nanoparticles to Reduce Cd Availability in Soils: Effects on Germination and Early Growth of Sinapis alba L."

_plants, 2023, doi:10.3390/plants12040801_

Round 1
Reviewer 1 Report
1. This study is well-organized and the writing looks acceptable, but it still needs revision before publishing.
2. The data in table 4 may have problems, why does the percentage of GI in untreated soil 0 still have SE of 1.41, it means that there is a negative percentage in this treatment. To me, it's very difficult to understand. Similar problems were found in several treatments in this table.
3. How many soil samples were conducted to analyze for general soil characteristics in table 1? How to ensure that this part of the data can be representative? Please describe soil sample preparation more in detail in the section of M&M.
4. Why did the authors select the three nanoparticles, including hydroxyapatite, maghemite, and zero-valent iron at specific doses in this study? Why not test other types of nanoparticles and other doses?
5. Is it possible to monitor the long-term effects of the plants on their morphology, growth, and development?
6. Keywords: "availability" is too general
7. Keywords: " early growth"; Do you mean "seedling growth"?
8. L10: "Cadmium" change to "Cadmium (Cd)"
9. L14: "Sinapis alba" should be in italics and add "L." in upright letters, and when it appears in the following text should be written as "S. alba".
10. L20-22: This part should be presented more conservatively because the authors only analyzed the performance of germination and seedling growth. It also needs to modify the related description in the parts of the results and discussion.
11. Rewrite similar sentences between the abstract and the conclusion.
12. The background information about the three types of nanoparticles used in this study is too weak including in the introduction and discussion.
Author Response
Reviewer 1
Comments and Suggestions for Authors
- This study is well-organized and the writing looks acceptable, but it still needs revision before publishing.
Thank you very much for your suggestions. We have revised the manuscript in deep.
- The data in table 4 may have problems, why does the percentage of GI in untreated soil 0 still have SE of 1.41, it means that there is a negative percentage in this treatment. To me, it's very difficult to understand. Similar problems were found in several treatments in this table.
Germination inhibition (GI) and/or root growth inhibition (RI) are defined as:
Where:
SRC is the average seed germination or root length for plants in the control soil (without Cd and nanoparticles)
SRS is the average seed germination or root length in the soils treated with Cd and/or nanoparticles.
To calculate GI and/or RI of the control soil, we considered the average germination in each of the three replicates of the plates, so being an average has an error. Likewise, the length of the roots also has a standard deviation in each plate, so when applying the formula, it is necessary to consider the propagation of errors.
On the other hand, the appearance of negative values in table 4 is feasible since negative values indicate the germination stimulation concerning the control soil.
However, the discussion in table 4 has been thoroughly revised, re-explained, and improved to make it much more understandable to the reader.
- How many soil samples were conducted to analyze for general soil characteristics in table 1? How to ensure that this part of the data can be representative? Please describe soil sample preparation more in detail in the section of M&M.
Thank you very much for your comment. We have described the soil sample preparation more in detail in section 3.1 Soil sampling and characterization. Besides, we have classified the soil to improve the understanding of the results with the type of soil used, according to IUSS Working Group. WRB World Reference Base for Soil Resources 2015. International Soil Classification System for Naming Soils and Creating Legends for Soil Maps; World Soil Resources Reports No 106, FAO, Rome; ISBN 9789251083697.
- Why did the authors select the three nanoparticles, including hydroxyapatite, maghemite, and zero-valent iron at specific doses in this study? Why not test other types of nanoparticles and other doses?
As we indicated in the introduction, phosphate-based, Fe oxides and zero Fe nanoparticles are among the most widely used in the environmental field for soil and water remediation. At present, their use has spread considerably, also covering agriculture. Therefore, it is necessary to obtain and expand information on the benefits and possible harmful effects of their use. For this reason, in this study, we have selected some of the most common phosphate-based nanoparticles, iron oxide and zero-valent iron nanoparticles in environmental studies in soils and waters. Several previous studies have evaluated the remediation potential of these nanoparticles in soils and waters contaminated with metals, such as those reported in references 38-40 and 42-45.
The nanoparticles and the dose applied were selected and adjusted based on similar doses used in previous papers.
Arenas-Lago, D., Abreu, M. M., Andrade Couce, L., & Vega, F. A. (2019). Is nanoremediation an effective tool to reduce the bioavailable As, Pb and Sb contents in mine soils from iberian pyrite belt? Catena, 176, 362-371. doi:10.1016/j.catena.2019.01.038
Nanoparticles: HANPs, MNPs and hematite nanoparticles. Dose applied: 5 % (w/w) (ratio NPs:soil)
Arenas-Lago, D., Rodríguez-Seijo, A., Lago-Vila, M., Couce, L. A., & Vega, F. A. (2016). Using Ca3(PO4)2 nanoparticles to reduce metal mobility in shooting range soils. Science of the Total Environment, 571, 1136-1146. doi:10.1016/j.scitotenv.2016.07.108
Nanoparticles: Tricalcium phosphate nanoparticles. Dose applied: 5 % (w/w).
Lago-Vila, M., Rodríguez-Seijo, A., Vega, F. A., & Arenas-Lago, D. (2019). Phytotoxicity assays with hydroxyapatite nanoparticles lead the way to recover firing range soils. Science of the Total Environment, 690, 1151-1161. doi:10.1016/j.scitotenv.2019.06.496
Nanoparticles: HANPs. Dose applied: 5 % (w/w).
Rodríguez-Seijo, A., Vega, F. A., & Arenas-Lago, D. (2020). Assessment of iron-based and calcium-phosphate nanomaterials for immobilisation of potentially toxic elements in soils from a shooting range berm. Journal of Environmental Management, 267 doi:10.1016/j.jenvman.2020.110640
Nanoparticles: HANPs, hematite nanoparticles. Dose applied: 5 % (w/w).
In previous similar studies the dose of nanoparticles applied was 5% (w/w). A reduction of the applied dose to 3% (w/w) was considered in order to reduce costs, and limit possible negative effects of nanoparticles on S. alba, aiming at a high capacity to reduce the available Cd content.
All, these references are cited in the manuscript.
- Is it possible to monitor the long-term effects of the plants on their morphology, growth, and development?
Thank you very much for your question. Yes, it is one of our priorities for the next studies. Still, we consider it essential to know the early effects of nanoparticles and Cd in germination and the first stages of the species' growth since it is one of the most vulnerable stages of the cycle of plant species in general. If the species cannot germinate and grow during the early phenological stages, its long-term development is not viable.
- Keywords: "availability" is too general
Thank you very much for your suggestion. We have deleted the keyword.
- Keywords: " early growth"; Do you mean "seedling growth"?
Thank you very much for your suggestion. We have replaced the keyword.
- L10: "Cadmium" change to "Cadmium (Cd)"
Done
- L14: "Sinapis alba" should be in italics and add "L." in upright letters, and when it appears in the following text should be written as "S. alba".
Done
- L20-22: This part should be presented more conservatively because the authors only analyzed the performance of germination and seedling growth. It also needs to modify the related description in the parts of the results and discussion.
Thank you very much. We have modified the text according to your suggestions.
- Rewrite similar sentences between the abstract and the conclusion.
We have revised and changed the abstract and the conclusion in deep.
- The background information about the three types of nanoparticles used in this study is too weak including in the introduction and discussion.
Thank you very much for your comment. We included similar studies (see point 4 of these comments) where these nanoparticles were used for immobilizing metals and other hazardous potential elements. Besides, we have added some references to the discussion section to enhance the founds.

Reviewer 2 Report
The study focused on restoration of soil Cd contamination using soil ameliorations and plant. The content falls within the scope of the journal and provided some new knowledge to the community, which make it worth to publish after a major revision. My concern with the present version is listed below and indicted in the marked version of the manuscript as in attached.
1. The language needs further revision, including grammar, format, etc.
2. The organization of the figures need further revision to make them more concise and focusing on the key findings.
3. The tables need more clear legend, and data in the table should be presented in consistent format.
4. Abstract need some results supported by data.
5. Other comments are included in the attachment.

Author Response
Reviewer 2
- The language needs further revision, including grammar, format, etc.
Thank you very much for your suggestion. The manuscript has been revised in deep by a native English speaker.
- The organization of the figures need further revision to make them more concise and focusing on the key findings.
We have reduced the number of figures to two. We have merged Figures 1-4 in Figure 1.
- The tables need more clear legend, and data in the table should be presented in consistent format.
We have changed the legends, and the figures were adjusted to be consistent.
- Abstract need some results supported by data.
We added some data to the abstract.
- Other comments are included in the attachment.
For table 1. Texture has no units. The pH is the –log[H+] and the unit for [H+] are mol L-1, but pH is expressed without units. The other parameters have units, but they were grouped. For example: Ca2+, K+, Mg, 2+ Na+, Al3+ and ECEC à cmol(+)kg-1
We have revised and corrected the other comments included in the attachment.

Round 2
Reviewer 1 Report
I don't have further questions.
Author Response
Thank you very much. We have added numerical data to the abstract to complete your suggestions.